# “My Life during the Lockdown”: Emotional Experiences of European Adolescents during the COVID-19 Crisis

**DOI:** 10.3390/ijerph18147638

**Published:** 2021-07-18

**Authors:** Alberto Forte, Massimiliano Orri, Martina Brandizzi, Cecilia Iannaco, Paola Venturini, Daniela Liberato, Claudia Battaglia, Isabel Nöthen-Garunja, Maria Vulcan, Asja Brusìc, Lauro Quadrana, Olivia Cox, Sara Fabbri, Elena Monducci

**Affiliations:** 1European Network for Psychodynamic Psychiatry, 00154 Rome, Italy; cecilia.iannaco@netforpp.eu (C.I.); paola.venturini@aslroma1.it (P.V.); daniela.liberato@netforpp.eu (D.L.); claudia.battaglia@uniroma1.it (C.B.); isabel.noethengarunja@netforpp.eu (I.N.-G.); 2Department of Psychiatry and Substance Abuse, ASL Roma 5, 00019 Rome, Italy; 3McGill Group for Suicide Studies, Douglas Mental Health University Institute, Department of Psychiatry, McGill University, Montreal, QC H3A 0G4, Canada; massimiliano.orri@mail.mcgill.ca; 4Bordeaux Population Health Research Centre, Inserm U1219, Université de Bordeaux, 30072 Bordeaux, France; 5Department of Psychiatry, ASL Roma 1, 00193 Rome, Italy; martina.brandizzi@aslroma1.it; 6Policlinico Umberto I, 00185 Rome, Italy; lauro.quadrana@uniroma1.it; 7Timișoara 2021—European Capital of Culture Association, 300057 Timișoara, Romania; maria.vulcan@timisoara2021.ro; 8Rijeka 2020—European Capital of Culture, Croatian Cultural Centre, 51000 Rijeka, Croatia; Asja@hkd-rijeka.hr; 9Department of Neurosciences and Mental Health, Section of Child and Adolescent Neuropsychiatry, Sapienza University of Rome, 00185 Rome, Italy; 10Faculty of Medicine and Psychology, Sapienza University, 00185 Rome, Italy; ollicox93@gmail.com (O.C.); sarafabbri92@gmail.com (S.F.); 11Department of Human Neurosciences, Sapienza University, 00185 Rome, Italy; elena.monducci@uniroma1.it

**Keywords:** adolescents, lockdown, emotional reactions, school-students

## Abstract

This study investigates, using an online self-report questionnaire, adolescents’ emotional reactions during the lockdown in a sample of 2105 secondary school students (aged 14–19) in Italy, Romania, and Croatia. We used a self-reported online questionnaire (answers on a 5-point scale or binary), composed of 73 questions investigating the opinions, feelings, and emotions of teenagers, along with sociodemographic information and measures of the exposure to lockdown. The survey was conducted online through a web platform in Italy (between 27 April and 15 June 2020), Romania, and Croatia (3 June and 2 July 2020). Students aged >14 years, living in a small flat, and not spending time outside were more likely to report anger, sadness, boredom/emptiness, and anxiety. Boys were significantly less likely than girls to report all measured emotional reactions. Those who lost someone from COVID-19 were more than twice as likely to experience anger compared to those who did not. Our findings may help identifying adolescents more likely to report negative emotional reactions during the COVID-19 pandemic and inform public health strategies for improving mental health among adolescents during/after the COVID-19 crisis.

## 1. Introduction

The COVID-19 epidemic started in China in December 2019 and rapidly spread worldwide [1]. To respond to the public health crisis, many countries introduced severe lockdown measures such as school closure, social distancing, interruption of sports activities, and quarantine/isolation [1]. These restrictive measures may have serious psychological consequences in young individuals [2], especially among the more vulnerable. 

Moreover, according to UNESCO, schools were closed in 188 countries between April and May 2019, leaving about 1.5 billion students out of the school system, representing 60% of the world student population [3]. School closures and decreased educational opportunities due to the pandemic might have a great impact on youths’ mental health [4], exposing young individuals to a higher risk of physical and/or sexual violence at home and, together with the economic damage caused by the crisis, may lead to increased mental health issues and suicide rates [5]. Furthermore, a recent systematic review of the literature showed a strong association between isolation/loneliness and depression in children and adolescents [6]. Additionally, schools play a fundamental role in the prevention and early recognition of mental disorders [7], especially for those with social and economic disadvantages [8], adding to the potential negative impact of COVID-19 on youth mental health. 

Recent studies have shown increased levels of depressive, anxiety, and post-traumatic stress symptoms among adolescents in the context of the COVID-19 pandemic [9], particularly among girls, senior high school/college students, and those with economic disadvantages [6,10,11,12,13,14]. However, only a few of them focused on the emotional reactions of adolescents during the COVID-19 outbreak [15] and on psychological distress [16,17,18]. Understanding the emotional reactions of secondary school students during the lockdown might help us to identify students at risk of psychological distress and inform preventive actions. In this study, we developed an online survey involving secondary school students from three European countries (Italy, Croatia, and Romania). The aim was to investigate the associations between several sociodemographic and lockdown-related factors with several emotional reactions (anger, anxiety, sadness, and boredom/emptiness).

## 2. Materials and Methods

### 2.1. Study Design

An international cross-sectional study was performed in three European countries (Italy, Romania, and Croatia) based on a questionnaire developed by the Department of Human Neurosciences/Section for Child and Adolescent Psychiatry, Sapienza University (Rome, Italy) and distributed by The European Network for Psychodynamic Psychiatry (Netforpp Europa, Rome, Italy) a non-profit organization, during and immediately after the lockdown.

### 2.2. Recruitment Procedure 

As described elsewhere [19] the survey was conducted online through a web platform (SurveyMonkey^®^) in Italy (between 27 April and 15 June 2020), Romania, and Croatia (between 3 June and 2 July 2020). Data collection was coordinated by Netforpp Europa in Italy and the associations Timisoara 2021 in Romania and Hrvatski Kulturi Dom in Croatia. The survey was conducted in a population of secondary school students as an extension of a previous project on mental health literacy, which had been conceived and implemented by Netforpp Europa, in collaboration with the Department of School Services of Rome (Italy), in several high schools in the cities of Rome and Florence in January 2020. Schools were recruited based on a previous EU funded project “*Mai Più Memory against Inhumanity: People with Mental Disorders under Totalitarian Regimes in Europe*”, which involved several schools in different European countries [20].

The Croatian coordinator collaborated with the Departments of Education and Schooling of the City of Rijeka and Primorsko-goranska County, which distributed the questionnaire among schools in Rijeka and its region. In order to obtain a nationwide coverage and adequate distribution of age groups, the coordinator in Romania collaborated with the “Europe Direct” network of information centers in Arad and Bucharest, the Timis Sibiu County School Inspectorates and several media partners. About half (*n* = 508) of the total number of 1004 responses in Romania was collected in the “Elena Ghiba Birta” National College in Arad. The other half of responses came from schools in Timișoara, Sibiu, and Resifa.

Every school was contacted 2 weeks before the survey started and the study protocol was outlined in detail to head and class teachers. In Italy, approximately 7500 students were invited to participate (response rate 12.3%). Each participant of the questionnaire remained anonymous and respondents’ IP addresses could not be disclosed. Participation was entirely voluntary and without any compensation. Participants over the age of 18 years gave their voluntary informed consent to participate in the research before taking part in the study. For students under 18 years of age, participating schools entered the questionnaire in the school’s electronic parental platform, together with a cover letter in which all the information on the study protocol and the survey’s objectives were given. Schools included in the study uploaded the questionnaire to the school electronic parental platform along with a cover letter explaining in detail the study protocol. The letter contained the purpose of the study, the study procedure, and information about data protection and privacy. The school electronic parenting platform is an electronic classroom-board that can only be accessed by parents with a personal password. Only after parental acknowledgment and consent were their children able to complete the questionnaire.

### 2.3. Measures

“My life during lockdown” is a self-reported online questionnaire, which is composed of 73 questions investigating the opinions, feelings, and emotions of teenagers, along with sociodemographic information and measures of the exposure to lockdown. Types of questions differed within the survey, and were either on a 5-point Likert scale or binary questions. The questionnaire was completed anonymously, and the respondents’ IP address was hidden. The questionnaire was developed in Italian and translated into the Romanian, Croatian, and English languages using a translation and back-translation procedure. Measured variables are listed in Table 1. We classified municipalities into three degrees of urbanization, according to EUROSTAT [21]; metropolitan areas (cities/large urban areas), medium-size urban areas (towns and suburbs/small urban areas), and rural areas. Parents socioeconomic status (SES) were classified according their occupation into low (i.e., (unskilled manual, non-manual low, farmer, fisher, retired/social welfare, or non-workers) vs. non-low (i.e., non-manual high, employee, self-employed and skilled worker, skilled manual, professional/managerial, or technical/skilled) [22]. Housing characteristics considered were: size of the house/apartment (> or <60 m^2^), possibility of spending time outside, and whether it was possible to have privacy in the apartment during the lockdown (binary questions). We also asked about COVID-19-related information (“Has somebody important to you contracted SARS-CoV2?” “Have you experienced the loss of a loved one because of SARS-CoV2?”). Moreover, we asked whether parents were still working during the lockdown (answers: yes, no but they kept their job, no and their have lost their job), and whether they were worrying for the economic situation (“Are you worried about economic problems during this period?”). We also included questions about: relationships (“Has your relationship with your parents/friends/partner changed during the lockdown?”, answered as no, yes negatively, yes positively); social media (“Has your use of social media increased during the lockdown?”, answered on a 5-point scale ranging from Not at all to A lot); the importance of family or professional support (“Do you think that being with your family helped you to overcome this period?”, “Have you ever thought it would help you to talk to a professional about your feelings?”); and the positive impact of the lockdown (“Did you discover new interests/hobbies or talents during this period?”, “Do you think that you have spent this period of time in a productive and creative way?”). Finally, students’ emotional reactions were assessed by asking students if they were experiencing feelings of anxiety, sadness, anger, and persistent boredom/emptiness during the lockdown period (binary questions). 

### 2.4. Data Analysis

The study variables were described using mean and standard deviation (for continuous variables) and count and percentage (for categorical variables). First, descriptive statistics were estimated in the whole sample and by country, and the comparison across countries was performed using t-tests and chi-square tests. Second, associations of sociodemographic and lockdown-related variables with the four measured emotional reactions were estimated using univariable logistic regressions. Third, the variables that were associated with the emotional reactions at *p* < 0.05 were entered in multivariable logistic regression models to estimate their independent associations with emotional reactions. All tests were 2-tailed and considered statistically significant at *p* < 0.05.

### 2.5. Ethical Issues

All subjects gave their written informed consent for inclusion before they participated in the study. Participation was voluntary and without compensation. The study was conducted in accordance with the Declaration of Helsinki. Moreover, the study followed the privacy recommendation released by the Italian Ministry of Education, University, and Research (MIUR) (https://www.miur.gov.it/privacy-tra-i-banchi-di-scuola (accessed on 15 July 2020)). The project received Institutional approval by the Municipality of Rome, Department of School and Education (institutional authorization number n.987 06/05/2019), and was considered in line with GDPR 2016/679 (General Data Protection Regulation).

## 3. Results

### 3.1. Participants’ Characteristics

A detailed description of the participants’ characteristics is presented in Table 1. Most of the participants were 14–16 years old (44.4%) or 16–18 years old (35.9%), and 68.7% were female. Of the respondents, 87.3% were from a densely populated area, and 86.7% reported living in an apartment of >60 m^2^. Regarding COVID-19-related variables, 6.7% indicated knowing people with COVID-19 (family members, relatives, and/or friends), while 1.5% reported knowing someone who had died from COVID-19. Several variables significantly differed among countries. Among sociodemographic variables, the degree of urbanization significantly differed among countries (*p* < 0.001); adolescents living in rural areas were mainly from Croatia and Romania, and none were from Italy (26.6%, 5.8%, and 0%, respectively), while Italian respondents were mainly from a densely populated area (97.5%). Additionally, housing variables significantly differed among countries for both the house size and the time spent outside: Italian respondents were more likely to live in larger houses (*p* < 0.001) and less likely to spend time outside (*p* < 0.001). Moreover, the fear of getting infected with COVID-19 was significantly higher among Italian adolescents compared to Romanian and Croatians (50.5%, 46.8%, and 27.7%, respectively, *p* < 0.001).

### 3.2. Univariable Analyses

Boredom/emptiness was the most frequently reported emotional reaction (*n* = 1504, 71.7%) followed by sadness (*n* = 1062, 50.5%), anxiety (*n* = 786, 37.3%), and anger (*n* = 698, 33.2%). Italian adolescents were more likely to report boredom/emptiness, anxiety, and sadness (*p* < 0.001), whereas no significant difference between countries emerged for anger. Among sociodemographic variables, we found that age was significantly associated with all outcomes; emotional reactions were more frequently reported by adolescents aged more than 16 years and 14–16 years, compared with those younger than 14 years (Table 2). Boys were less likely than girls to report all measured emotional reactions (anger: OR: 0.64; 95% CI: 0.52–0.79; sadness OR: 0.35; 95% CI: 0.28–0.42; boredom/emptiness: OR: 0.56; 95% CI: 0.46–0.68; and anxiety: OR: 0.42; 95% CI: 0.34–0.52). Similarly, adolescents living in rural areas were less likely to report all emotional reactions than those living in urban areas (anger: OR: 0.51; 95% CI: 0.31–0.83; sadness OR: 0.53; 95% CI: 0.34–0.83; boredom/emptiness: OR: 0.59; 95% CI: 0.37–0.94; and anxiety: OR: 0.28; 95% CI: 0.15–0.51). 

Having a small house was significantly associated with anger (OR: 1.3; 95% CI: 1.00–1.69), while not spending time outside was significantly associated with anger, sadness, and boredom/emptiness (anger: OR: 1.4; 95% CI: 1.14–1.73; sadness OR: 1.27; 95% CI: 1.04–1.55; and boredom/emptiness: OR: 1.26; 95% CI: 1.03–1.54) (Table 2). Interestingly, several lockdown-related and COVID-19-related variables were associated with emotional reactions, and thus included in the multivariable analysis (Table 2).

### 3.3. Multivariable Analyses

Table 3 reports the results of the multivariable analyses including only variables that were significantly associated with the outcomes in the univariate analysis.

*Sociodemographic variables.* Among sociodemographic variables, we found that the likelihood of reporting anger (OR: 1.59; 95% CI: 1.19–2.11), sadness (OR: 1.67; 95% CI: 1.28–2.18), boredom/emptiness (OR: 1.41: 95% CI: 1.06–1.87), and anxiety (OR: 1.59: 95% CI: 1.18–2.13) were higher among adolescents aged 14–16 years and among those older than 16 years, compared to those aged <14 years, in line with the univariable analyses. Similarly, being a boy was still independently associated with a lower likelihood of reporting all emotional reactions (Table 3). Living in a rural area was associated with a decreased likelihood of experiencing anxiety (OR: 0.41; 95% CI: 0.21–0.78), but the association with anger, boredom/emptiness, and sadness/depression were no longer significant in the multivariable analysis.

Housing. Adolescents reporting not spending time outside their home during the lockdown were significantly more likely to experience anger (OR: 1.33; 95% CI: 1.06–1.66) and sadness (OR: 1.3; 95% CI: 1.04–1.63).

COVID-19-related variables. We found that adolescents who reported that a loved person had died from COVID had a more than two times higher risk of reporting feelings of anger (OR: 2.74; 95% CI: 1.29–5.81) compared with an adolescent who did not. Interestingly, strongly trusting the government was found to be significantly protective against experiencing anger (OR: 0.52; 95% CI: 0.35–0.76), sadness (OR: 0.62; 95% CI: 0.43–0.89), and boredom/emptiness (OR: 0.62; 95% CI: 0.42–0.91).

Relationships with parents and peers. We found that those reporting a negative impact of the lockdown on their relationships with friends were at higher risk of experiencing all emotional difficulties (e.g., OR for anxiety: 1.7; 95% CI: 1.26–2.30). Additionally, adolescents reporting a negative impact on their relationships with parents were more likely to experience sadness (OR: 1.55; 95% CI: 1.10–2.20).

Social media. We found a significantly increased likelihood of experiencing all emotional reactions among adolescents who reported increased use of social media (Table 3), especially boredom/emptiness (OR: 1.44; 95% CI: 1.32–1.57).

Support. We found that those who believed that family support was important during the lockdown were less likely to report both boredom/emptiness (OR: 0.66; 95% CI: 0.52–0.84) and anxiety (OR: 0.61; 95% CI: 0.48–0.78) compared to those who did not believe so. In contrast, those who considered external support as important were more likely to report all emotional difficulties except anger. Notably, those reporting to be in psychotherapy were more likely to report boredom/emptiness (OR: 1.72; 95% CI: 1.08–2.74).

The positive impact of the lockdown. Adolescents who reported to spend time creatively were significantly less likely to experience anger (OR: 0.78; 95% CI: 0.64–0.96), sadness (OR: 0.66; 95% CI: 0.54–0.81), and boredom/emptiness (OR: 0.51; 95% CI: 0.41–0.63) than those who did not spend time creatively.

## 4. Discussion

The present study reports the findings from a survey on the emotional reactions of a large sample of European secondary school students during the COVID-19 crisis. We found that the likelihood of experiencing anger, sadness, boredom/emptiness, and anxiety was higher among oldest (>14 years) and female adolescents, and related to housing characteristics and time spent outside. This is partly consistent with other studies where the female gender was found to be related to higher levels of psychological distress [23,24,25]. Additionally, recent studies on Chinese adolescents showed that older girls (15–18 years) were more likely to present depressive/anxious symptoms [11,26]. This might be, in part, explained by the fact that girls are generally more prone to internalizing-spectrum symptoms [27], while boys might be more likely to show externalizing behaviors and underreport internalizing emotions [28]. This gender difference might inform school-based preventive, gender-targeted interventions; it might also suggest that the detection of internalizing emotional difficulties might be underestimated among boys, who are at higher risk of developmental difficulties and negative later mental health outcomes than girls [29], including a higher risk of suicide [30,31]. Notably, even if boys were often less likely to express emotional distress, this might be related to societal and cultural constraints resulting in underreporting of emotional difficulties [29]. This might also suggest a need for addressing healthy masculinities and gender equality in emotional expressions [32] 

We also found several protective factors. Living in a rural area was protective against experiencing anxiety, while spending time creatively during the lockdown was significantly protective of experiencing anger, sadness, and boredom/emptiness. Consistently, living in an urban area was already found to be a risk factor for experiencing anxiety among college students [33]. Thus, societal disparities, such as housing characteristics, might exacerbate the adverse emotional effects of the COVID-19 pandemic and have an impact on the emotional reactions of adolescents [34]. Our findings expand the knowledge on the protective effect of daily routine and positive reframing [35], which have been found to be protective factors against perceived stress and emotional difficulties [24]. This was also found in a previous Italian survey, which reported that reconstructing a sort of daily “agenda” during the lockdown helped the overall emotional balance of children [36]. 

Results from the present survey provide important insights into adolescents’ emotional reaction after losing someone from COVID-19; we found that young individuals who lost a loved one from COVID-19 were at twice the risk of experiencing anger compared to those who did not (OR: 2.74; 95% CI: 1.29–5.81). The feeling of anger was already found to be a common emotional reaction among adolescents during lockdown [24], but our study expands the knowledge by suggesting that it is important to address the feeling of anger among young people who have lost someone from COVID-19. Indeed, reacting with anger might suggest that COVID-19 related grief among adolescents is experienced with a sense of injustice, which is typical of conflict-related trauma [37]. Future research is necessary to better understand the complexity of grief reactions among adolescents who lost someone from COVID-19, as this may inform preventive and therapeutic interventions. A previous study also reported that anger was found as a traumatic reaction among the general population of the Czech Republic, and that this was related to mass media pessimism [38]. Thus, our findings also suggest that future research studies are needed to clarify the association between mass media reporting and adolescent emotional reactions.

Interestingly, we also found that increased use of social media was significantly associated with all negative emotional reactions investigated, consistently with previous studies [10,39]. However, the direction of this association is difficult to interpret, as adolescents experiencing emotional difficulties might use social media more frequently [40]. Further studies are needed, aimed at understanding the role of social media in identifying youth in need of help, who are more likely to report negative emotional reactions.

Our findings also highlight the importance of the relationship with peers; those reporting a negative impact of the lockdown on their relationship with peers were at higher risk of experiencing negative emotional reactions, particularly more anxiety. This finding is in line with previous evidence highlighting the important role of peer relationships in the development of anxiety among adolescents [41,42]. Moreover, this suggests the importance of establishing peer support networks, either facilitated by peers or by professional interventions [43,44,45].

Notably, our findings also showed that trusting government decision-making could be considered a protective factor against negative emotional reactions among young students; strongly trusting the government was found to be significantly protective against experiencing anger, sadness, and boredom/emptiness. Previous findings also suggested that improving knowledge and positive attitudes toward the crisis among young people might enhance their resilience and reduce the risk of the psychological burden of restrictive measures [26].

## 5. Limitations

The present findings should be interpreted in light of several limitations. First, the cross-sectional design; emotional difficulties were measured at the same assessment, during the lockdown, thus the directions of the associations described are uncertain. Moreover, students were not randomly selected, and this might limit the generalizability to the entire population. Furthermore, the overall mental health status of the respondents is not known prior to the pandemic, so self-selection to complete the survey and participate may be taken into account in interpreting the results. Finally, in the absence of pre-pandemic data, we cannot know if several of the reported associations (e.g., between social media and anxiety) are specifically related to the current COVID-19 crisis or are more general associations that we would have observed independently from the crisis. 

## 6. Conclusions

The present survey demonstrated that the risk of experiencing anger, sadness, boredom/emptiness, and anxiety was higher among older adolescents, females, and adolescents living in a small flat, not spending time outside, and reporting increased use of social media. Losing a loved one from COVID-19 was specifically associated with anger among affected adolescents, suggesting a specific reaction to such a tragic event. Nonetheless, several protective factors were identified, such as spending time creatively during the lockdown and trusting the government’s decisions. The present findings might help to identify adolescents more likely to report negative emotional reactions during the COVID-19 pandemic and inform policymakers and future public health strategies on improving mental health among adolescents. Additionally, the present study might inform future research on school-based preventive interventions, suggesting that improving trust in public health policies, social connectedness, as well as improving knowledge and positive attitudes toward the health crisis, might enhance resilience and reduce the risk of psychological burden among school students.

## Figures and Tables

**Table 1 ijerph-18-07638-t001:** Characteristics of the Sample.

	Category	Whole Sample (2105)	Italy (928)	Romania (1004)	Croatia (173)	*p* Values
Sociodemographic variables						
Age	<14	415 (19.7)	140 (15.1)	232 (23.1)	43 (24.9)	<0.001
	>16	755 (35.9)	335 (36.1)	367 (36.6)	53 (30.6)	
	14–16	935 (44.4)	453 (48.8)	405 (40.3)	77 (44.5)	
Gender	Girls	1446 (68.7)	694 (74.8)	638 (63.5)	114 (65.9)	<0.001
	Boys	659 (31.3)	234 (25.2)	366 (36.5)	59 (34.1)	
Degree of urbanization	Densely populated area	1837 (87.3)	905 (97.5)	848 (84.5)	84 (48.6)	<0.001
	Intermediate density areas	164 (7.8)	23 (2.5)	98 (9.8)	43 (24.9)	
	Rural	104 (4.9)	0 (0.0)	58 (5.8)	46 (26.6)	
Siblings	No	590 (28.0)	206 (22.2)	342 (34.1)	42 (24.3)	<0.001
	One	1104 (52.4)	530 (57.1)	481 (47.9)	93 (53.8)	
	More than one	411 (19.5)	192 (20.7)	181 (18.0)	38 (22.0)	
Mother low SES	Yes	1318 (62.6)	676 (72.8)	545 (54.3)	97 (56.1)	<0.001
	No	787 (37.4)	252 (27.2)	459 (45.7)	76 (43.9)	
Father low SES	Yes	526 (25.0)	95 (10.2)	335 (33.4)	96 (55.5)	<0.001
	No	1579 (75.0)	833 (89.8)	669 (66.6)	77 (44.5)	
Housing						
House surface	>60 m^2^	1825 (86.7)	848 (91.4)	829 (82.6)	148 (85.5)	<0.001
	<60 m^2^	280 (13.3)	80 (8.6)	175 (17.4)	25 (14.5)	
Time outside home	Yes	1590 (75.5)	648 (69.8)	795 (79.2)	147 (85.0)	<0.001
	No	515 (24.5)	280 (30.2)	209 (20.8)	26 (15.0)	
Privacy	Yes	1778 (84.5)	775 (83.5)	847 (84.4)	156 (90.2)	0.084
	No	327 (15.5)	153 (16.5)	157 (15.6)	17 (9.8)	
COVID-19-related						
Loved one with COVID	Yes	141 (6.7)	80 (8.6)	56 (5.6)	5 (2.9)	0.003
	No	1964 (93.3)	848 (91.4)	948 (94.4)	168 (97.1)	
Loved one died of COVID	Yes	32 (1.5)	23 (2.5)	9 (0.9)	0 (0.0)	0.004
	No	2073 (98.5)	905 (97.5)	995 (99.1)	173 (100.0)	
Fear of getting COVID	Yes	987 (46.9)	469 (50.5)	470 (46.8)	48 (27.7)	<0.001
	No	1118 (53.1)	459 (49.5)	534 (53.2)	125 (72.3)	
Trust in the government	No	375 (17.8)	158 (17.0)	182 (18.1)	35 (20.2)	<0.001
	Yes, enough	1492 (70.9)	675 (72.7)	715 (71.2)	102 (59.0)	
	Yes, fully	238 (11.3)	95 (10.2)	107 (10.7)	36 (20.8)	
Job/economy						
Parents currently working	Yes	1592 (75.6)	689 (74.2)	774 (77.1)	129 (74.6)	0.014
	No, kept job	457 (21.7)	216 (23.3)	208 (20.7)	33 (19.1)	
	No, lost job	56 (2.7)	23 (2.5)	22 (2.2)	11 (6.4)	
Worries about money *		3.58 (1.08)	3.31 (1.07)	3.78 (1.03)	3.91 (1.07)	<0.001
Relationships						
Changed relationship parents	No	509 (24.2)	175 (18.9)	291 (29.0)	43 (24.9)	<0.001
	Positive	1298 (61.7)	585 (63.0)	597 (59.5)	116 (67.1)	
	Negative	298 (14.2)	168 (18.1)	116 (11.6)	14 (8.1)	
Changed relationship friends	No	474 (22.5)	179 (19.3)	258 (25.7)	37 (21.4)	<0.001
	Positive	1088 (51.7)	525 (56.6)	462 (46.0)	101 (58.4)	
	Negative	543 (25.8)	224 (24.1)	284 (28.3)	35 (20.2)	
Changed relationship partner	No/no partner	1287 (61.1)	594 (64.0)	669 (66.6)	24 (13.9)	<0.001
	Positive	396 (18.8)	177 (19.1)	205 (20.4)	14 (8.1)	
	Negative	422 (20.0)	157 (16.9)	130 (12.9)	135 (78.0)	
Social media						
Increased social media use *		2.55 (1.20)	2.33 (1.13)	2.69 (1.21)	2.87 (1.32)	<0.001
Support						
Helpful family support	Yes	1568 (74.5)	681 (73.4)	749 (74.6)	138 (79.8)	0.208
	No	537 (25.5)	247 (26.6)	255 (25.4)	35 (20.2)	
Professional support	Already in therapy	99 (4.7)	67 (7.2)	23 (2.3)	9 (5.2)	<0.001
	Yes	651 (30.9)	316 (34.1)	307 (30.6)	28 (16.2)	
	No	1355 (64.4)	545 (58.7)	674 (67.1)	136 (78.6)	
Positive impact						
New interests	Yes	1311 (62.3)	615 (66.3)	605 (60.3)	91 (52.6)	0.001
	No	794 (37.7)	313 (33.7)	399 (39.7)	82 (47.4)	
Creative time	Yes	1310 (62.2)	599 (64.5)	605 (60.3)	106 (61.3)	0.146
	No	795 (37.8)	329 (35.5)	399 (39.7)	67 (38.7)	

All variables are described as *n* (%), except for the variables with an * that are described as mean (SD).

**Table 2 ijerph-18-07638-t002:** Univariable Analysis. Logistic regression models estimating the association between each variable in column 1 and the outcomes in the last four columns (Anger, Sadness, Boredom/emptiness, and Anxiety). All analyses are adjusted for country. Boredom/emptiness has been categorized as follows: always, often = 1; never, rarely, sometimes = 0. Statistically significant variables are in bold.

Variable	Category	Anger OR (95% CI)	Sadness OR (95% CI)	Boredom/Emptiness OR (95% CI)	Anxiety OR (95% CI)
Sociodemographic					
Age	>16 vs. <14	**2.05** (**1.55–2.69**)	**2.19** (**1.71–2.8**)	**2.32** (**1.79–3.01**)	**2.92** (**2.22–3.85**)
	14–16 vs. <14	**1.85** (**1.42–2.42**)	**1.96** (**1.54–2.49**)	**1.83** (**1.42–2.35**)	**2.14** (**1.63–2.8**)
Gender	Boy	**0.64** (**0.52–0.79**)	**0.35** (**0.28–0.42**)	**0.56** (**0.46–0.68**)	**0.42** (**0.34–0.52**)
Urbanization	Intermediate density areas	0.8 (0.55–1.14)	1.04 (0.75–1.45)	1.09 (0.78–1.53)	**0.63** (**0.43–0.91**)
	Rural	**0.51** (**0.31–0.83**)	**0.53** (**0.34–0.83**)	**0.59** (**0.37–0.94**)	**0.28** (**0.15–0.51**)
Siblings	Yes one	1.06 (0.86–1.32)	1.04 (0.85–1.28)	1.11 (0.9–1.36)	1.02 (0.83–1.26)
	Yes, more than one	1.29 (0.99–1.69)	1.1 (0.85–1.41)	1.18 (0.91–1.52)	0.97 (0.75–1.27)
Mother low SES	Yes	0.93 (0.77–1.13)	0.88 (0.73–1.05)	**0.73** (**0.61–0.88**)	0.96 (0.8–1.16)
Father low SES	Yes	1.1 (0.88–1.38)	1.03 (0.84–1.27)	1.08 (0.87–1.34)	0.99 (0.79–1.24)
Housing					
House	<60 m^2^	**1.3** (**1.00–1.69**)	1.16 (0.9–1.49)	1.07 (0.83–1.39)	1.23 (0.94–1.59)
Time outside home	No	**1.4** (**1.14–1.73**)	**1.27** (**1.04–1.55**)	**1.26** (**1.03–1.54**)	1.21 (0.99–1.49)
Privacy	No	**1.58** (**1.24–2.02**)	1.24 (0.98–1.57)	**1.55** (**1.22–1.97**)	**1.34** (**1.05–1.7**)
COVID-19-related					
Loved one with COVID	Yes	1.41 (0.99–2)	1.33 (0.94–1.88)	**1.55** (**1.1–2.18**)	**1.66** (**1.17–2.34**)
Loved one died of COVID	Yes	**3.45** (**1.67–7.11**)	1.5 (0.73–3.08)	1.62 (0.8–3.29)	1.3 (0.64–2.62)
Fear of getting COVID	Yes	1.11 (0.92–1.33)	**1.46** (**1.22–1.73**)	1.13 (0.95–1.35)	**1.5** (**1.25–1.8**)
Trust in the government	Yes, enough	**0.63** (**0.5–0.8**)	0.8 (0.64–1.01)	**0.6** (**0.48–0.75**)	0.79 (0.63–1.00)
	Yes, fully	**0.37** (**0.25–0.53**)	**0.46** (**0.33–0.64**)	**0.37** (**0.26–0.52**)	**0.45** (**0.31–0.64**)
Job/economy					
Parent’s job	Still employed	0.99 (0.80–1.24)	1.20 (0.97–1.48)	1.04 (0.84–1.28)	1.03 (0.83–1.28)
	Unemployed	0.70 (0.38–1.28)	1.45 (0.84–2.49)	1.49 (0.87–2.55)	1.50 (0.87–2.58)
Worries about money		**1.31** (**1.2–1.43**)	**1.47** (**1.35–1.6**)	**1.51** (**1.39–1.65**)	**1.53** (**1.4–1.67**)
Relationships					
Changed relationship parents	Positive	0.87 (0.68–1.13)	0.99 (0.78–1.27)	0.95 (0.73–1.23)	1.07 (0.82–1.39)
	Negative	1.09 (0.77–1.53)	1.55 (1.1–2.2)	1.1 (0.77–1.56)	1.19 (0.84–1.7)
Changed relationship friends	Positive	1.26 (0.96–1.66)	1.09 (0.84–1.42)	1.04 (0.79–1.37)	1.14 (0.86–1.51)
	Negative	**1.55** (**1.15–2.09**)	**1.5** (**1.13–2**)	**1.4** (**1.04–1.88**)	**1.7** (**1.26–2.3**)
Changed relationship partner	Positive	0.93 (0.72–1.21)	1.16 (0.9–1.51)	1.07 (0.82–1.39)	1.17 (0.89–1.52)
	Negative	0.93 (0.7–1.23)	1.07 (0.81–1.41)	1.27 (0.95–1.68)	1.04 (0.78–1.39)
Social media					
Increased use		**1.35** (**1.24–1.46**)	**1.3** (**1.2–1.4**)	**1.63** (**1.5–1.76**)	**1.39** (**1.28–1.5**)
Support					
Helpful family support	Yes	**0.64** (**0.52–0.79**)	**0.75** (**0.61–0.91**)	**0.48** (**0.39–0.58**)	**0.5** (**0.41–0.61**)
Helpful external support	Yes	**1.52** (**1.24–1.85**)	**2.11** (**1.74–2.55**)	**2.22** (**1.84–2.7**)	**2.44** (**2.01–2.97**)
	Already in therapy	1.49 (0.98–2.28)	1.37 (0.91–2.07)	**2.28** (**1.5–3.46**)	**2.26** (**1.49–3.43**)
Positive impact					
New interests	Yes	0.94 (0.78–1.13)	0.97 (0.82–1.16)	**0.8** (**0.67–0.96**)	0.99 (0.83–1.2)
Creative time	Yes	**0.61** (**0.51–0.73**)	**0.59** (**0.5–0.71**)	**0.39** (**0.32–0.46**)	**0.51** (**0.42–0.61**)

**Table 3 ijerph-18-07638-t003:** Multivariable Analysis. Logistic regression models estimating the independent association between all the variables in column 1 (associated at *p* < 0.05 in the univariable analyses) and the outcomes in the last four columns; all models are also adjusted for country. Statistically significant variables are in bold.

Variable	Category	Anger OR (95% CI)	Sadness OR (95% CI)	Boredom/Emptiness OR (95% CI)	Anxiety OR (95% CI)
Sociodemographic					
Age	>16	**1.53** (**1.14–2.07**)	**1.55** (**1.17–2.05**)	**1.42** (**1.05–1.92**)	**1.78** (**1.31–2.43**)
	14–16	**1.59** (**1.19–2.11**)	**1.67** (**1.28–2.18**)	**1.41** (**1.06–1.87**)	**1.59** (**1.18–2.13**)
Gender	Male	**0.76** (**0.61–0.94**)	**0.4** (**0.33–0.5**)	**0.71** (**0.57–0.89**)	**0.55** (**0.44–0.69**)
Urbanization	Intermediate density areas	0.84 (0.57–1.23)	1.09 (0.76–1.57)	1.2 (0.82–1.76)	0.69 (0.46–1.04)
	Rural	0.72 (0.43–1.22)	0.74 (0.45–1.21)	0.95 (0.56–1.61)	**0.41** (**0.21–0.78**)
Mother low SES	Yes	-	-	**0.80** (**0.65–0.98**)	-
Housing					
House	<60 m^2^	1.07 (0.8–1.42)	-	-	-
Time outside home	No	**1.33** (**1.06–1.66**)	**1.3** (**1.04–1.63**)	1.17 (0.93–1.48)	-
Privacy	No	1.2 (0.91–1.57)	-	0.98 (0.74–1.3)	0.94 (0.71–1.24)
COVID-19-related					
Loved one with COVID	Yes	-	-	-	1.31 (0.9–1.91)
Loved one died of COVID	Yes	**2.74** (**1.29–5.81**)	-	1.11 (0.51–2.44)	-
Fear of getting COVID	Yes	-	**1.31** (**1.08–1.59**)	1.08 (0.89–1.32)	**1.46** (**1.19–1.79**)
Trust in the government	Yes, enough	**0.72** (**0.57–0.93**)	0.87 (0.67–1.12)	**0.71** (**0.55–0.93**)	0.95 (0.73–1.23)
	Yes, fully	**0.52** (**0.35–0.76**)	**0.62** (**0.43–0.89**)	**0.62** (**0.42–0.91**)	0.74 (0.5–1.11)
Job/economy					
Worries about money		**1.15** (**1.05–1.27**)	**1.28** (**1.17–1.41**)	**1.32** (**1.2–1.46**)	**1.32** (**1.2–1.46**)
Relationships					
Changed relationship parents	Positive	0.87 (0.68–1.12)	1 (0.79–1.28)	0.94 (0.73–1.22)	1.08 (0.83–1.4)
	Negative	1.1 (0.79–1.55)	**1.55** (**1.1–2.2**)	1.08 (0.76–1.54)	1.22 (0.86–1.72)
Changed relationship friends	Positive	1.27 (0.97–1.67)	1.08 (0.83–1.4)	1.04 (0.79–1.37)	1.14 (0.86–1.5)
	Negative	**1.55** (**1.16–2.09**)	**1.49** (**1.12–1.99**)	**1.38** (**1.03–1.86**)	**1.7** (**1.26–2.3**)
Changed relationship partner	Positive	0.93 (0.71–1.2)	1.16 (0.9–1.49)	1.07 (0.82–1.39)	1.14 (0.88–1.48)
	Negative	0.91 (0.69–1.21)	1.07 (0.81–1.41)	1.27 (0.96–1.69)	1.04 (0.78–1.38)
Social media					
Increased use		**1.23** (**1.13–1.34**)	**1.12** (**1.03–1.21**)	**1.44** (**1.32–1.57**)	**1.18** (**1.09–1.29**)
Support					
Helpful family support	Yes	0.89 (0.7–1.12)	1.02 (0.81–1.29)	**0.66** (**0.52–0.84**)	**0.61** (**0.48–0.78**)
Helpful external support	Yes	1.21 (0.98–1.5)	**1.49** (**1.2–1.84**)	**1.73** (**1.39–2.14**)	**1.72** (**1.39–2.13**)
	Already in therapy	1.08 (0.69–1.69)	0.89 (0.57–1.39)	**1.72** (**1.08–2.74**)	1.49 (0.95–2.33)
Positive impact					
New interests	Yes	-	-	0.93 (0.75–1.15)	-
Creative time	Yes	**0.78** (**0.64–0.96**)	**0.66** (**0.54–0.81**)	**0.51** (**0.41–0.63**)	**0.63** (**0.51–0.78**)

## Data Availability

The data that support the findings of this study are available on request from the corresponding author. The data are not publicly available due to privacy or ethical restrictions.

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
