# Peer review of "“My Life during the Lockdown”: Emotional Experiences of European Adolescents during the COVID-19 Crisis"

_ijerph, 2021, doi:10.3390/ijerph18147638_

Round 1
Reviewer 1 Report
Overall: Good study and highly relevant. Some sections can use additional details to help the reader understand the focus and methods.
Abstract: Good overall abstract and summary of study and findings. The abstract would benefit from additional information about the type of measurements (statistics) or question type (was it a multiple choice or Likert scale) as well as number of questions and timeline in which it was distributed.
Introduction:
Line 36 – Please clarify if the school closures simply made for home/distance education or if it was a lack of educational opportunities.
Materials and Methods:
Recruitment Procedure – how were the participants recruited? What methods were used to find or recruit participation in the study (specifically)? What are the details of the demographics of the population that this targeted (participant statistics as in age, ethnicity, number in each country, etc).
In the Measures section, please clarify the number and types of questions (multiple choice, Likert scale, etc) that the questionnaire provides to participants. There are some examples for specific questions, but it would be good to know the overall structure of the questions.
Ethical Issues – the method of participant recruitment should be earlier in the recruitment procedure.
Results: Clear and well-stated
Discussion: Line 230 – it might be useful to note here that males are often less likely to express emotional distress, but that does not mean they are less likely to experience it and that cultural/societal pressures may skey results in this way.
Limitations: Good discussion of limitations
Conclusions: Good conclusion of article and findings.
Author Response
Reviewer 1
Overall: Good study and highly relevant. Some sections can use additional details to help the reader understand the focus and methods.
Authors: we thank the reviewer for the positive feedback and the time he/she dedicated to giving us useful suggestions for improving the MS. We provided additional information in the method section as described below.
Abstract: Good overall abstract and summary of study and findings. The abstract would benefit from additional information about the type of measurements (statistics) or question type (was it a multiple choice or Likert scale) as well as a number of questions and timeline in which it was distributed.
Authors: according to your comments we added the above-mentioned information in the abstract section.
Introduction:
Line 36 – Please clarify if the school closures were simply made for home/distance education or if it was a lack of educational opportunities.
Authors: the mentioned study was conducted in China and there is no mention of distance education, thus we rephrased it as follows: School closures and decreased educational opportunities due to the pandemic might have a great impact on youths’ mental health
Materials and Methods:
Recruitment Procedure – how were the participants recruited? What methods were used to find or recruit participation in the study (specifically)? What are the details of the demographics of the population that this targeted (participant statistics as in age, ethnicity, number in each country, etc).
Authors: as suggested we outlined more in detail the recruitment procedure in each country in the method section. Sociodemographic variables are described in detail in Table1, where exact numbers related to each country can also be found.
In the Measures section, please clarify the number and types of questions (multiple choice, Likert scale, etc) that the questionnaire provides to participants. There are some examples for specific questions, but it would be good to know the overall structure of the questions.
Answers: we specified the above-mentioned information in the method section. Items had different response scales, i.e., either a 5-point Likert scale or binary, therefore we specified in the text the type of questions.
Ethical Issues – the method of participant recruitment should be earlier in the recruitment procedure.
Authors: we thank the reviewer for this suggestion. We moved the information on the recruitment procedure earlier in the manuscript.
Results: Clear and well-stated
Discussion: Line 230 – it might be useful to note here that males are often less likely to express emotional distress, but that does not mean they are less likely to experience it and that cultural/societal pressures may skey results in this way.
Authors: We appreciate that the reviewer pointed out this interesting aspect of gender differences in emotional expression among adolescents, which is in line with what we found in the present study. As argued by some authors (we cited Kramer, 2000, “The fragile male”), this is a key aspect of gender differences in emotional distress in adolescence. We also highlighted that, overall, negative outcomes and underreporting of mental health conditions among boys might be related to cultural and societal pressures.
Limitations: Good discussion of limitations
Conclusions: Good conclusion of article and findings
Authors: We appreciate the positive feedback from the reviewer
Reviewer 2 Report
This is a very important study, cataloguing the immense impact of COVID-19 on youth. While it will take some time to know of all the long term impacts, this study and others like it will be valuable later on to understand long term impacts knowing the more immediate experiences as we all are going through the pandemic.
A couple of suggestions to provide greater clarity:
1. Page 1, line 36. Moreover, according to UNESCO, schools were closed in 188 countries; instead of stating : have been closed
2. Why were Italy, Croatia, and Romania selected for the survey? Because of convenience? They represented on a continum the most to least impacted by COVID-19?
3. How many families/households received the survey? Is the 2,015 the final sample? Are you able to calculate and provide the response rate on the whole, by country?
4. Were there any psychometric analysis (i.e. reliability) done on any portion of the self-report questionnaire?
5. Limitations-- The overall mental health status of the respondents is not known prior to the Pandemic, so self-selection to complete the survey and participate may also may be a factor that must be taken into account in interpreting the results. Perhaps this could be integrated into the limitations section.
6. Conclusions: What recommendations would the authors have for social media to help youth identify, address/cope with the emotional distress from becoming mental illness symptoms, since youth turn to social media so much. Does social media have a role in helping identifying youth in need of help, who are more likely to report negative emotional reactions?
Additionally, do the the authors feel that their findings indicate importance of helping youth make connection as a way out of anger ,sadness ,etc. And,therefore, they would recommend that research on school-based preventative interventions should examine whether those intervention promote connection and connectedness, and what impact that has as protective factor against anger, sadness, boredom/emptiness, and anxiety as reported in this study. Should research and practice be recommended on gender specific programming (such as all girls or boys programming) as opposed to mixed-sex group programming given the findings of the study. Or would it be reasonable to recommend that, overall, social isolation is the major culprit here, and must be understood with new research and combatted with new programming.
Author Response
Reviewer 2
This is a very important study, cataloguing the immense impact of COVID-19 on youth. While it will take some time to know of all the long term impacts, this study and others like it will be valuable later on to understand long term impacts knowing the more immediate experiences as we all are going through the pandemic.
Authors: We thank the reviewer for highlighting the strengths of the paper, as well as for the time he/she dedicated to giving us useful suggestions for improving the MS.
A couple of suggestions to provide greater clarity:
- Page 1, line 36. Moreover, according to UNESCO, schools wereclosed in 188 countries; instead of stating : have been closed
Authors: we modified accordingly
- Why were Italy, Croatia, and Romania selected for the survey? Because of convenience? They represented on a continum the most to least impacted by COVID-19?
Authors: Countries were selected based on previous collaborations among the different research groups, the paragraph now read as follows: “Schools were recruited based on a previous EU funded project “Mai Più Memory Against Inhumanity: People with Mental Disorders under Totalitarian Regimes in Europe”, which involved several schools in different European countries [20].”
Despite we have no information to sort the three nations on the basis of COVID-19 impact, we believe that examining differences and similarities among countries that adopted different restrictive rules to manage the pandemic provides useful information on the impact of such restriction on adolescent emotions.
- How many families/households received the survey? Is the 2,015 the final sample? Are you able to calculate and provide the response rate on the whole, by country?
Authors: Response rate was not possible to be calculated as we could only really know the ratio between the students who could potentially access the questionnaire as outlined in the Methods section; due to the pandemic crisis, we could only ask the school heads/headteachers to put our questionnaire on their school homepage (accessed only by a personal password for students over 18 or by a password from parents for underage children). We then asked the teachers to inform their pupils during their online lessons. However, there might be a number of students who were not reached and didn’t receive the invitation. Thus, the response rate based on the number of students attending the schools involved might not be realistic.
- Were there any psychometric analysis (i.e. reliability) done on any portion of the self-report questionnaire?
Authors: No psychometric analysis was necessary because our constructs were measured using single items.
- Limitations-- The overall mental health status of the respondents is not known prior to the Pandemic, so self-selection to complete the survey and participate may also may be a factor that must be taken into account in interpreting the results. Perhaps this could be integrated into the limitations section.
Authors: This is a very pertinent comment, and we agree with the reviewer. As suggested, we integrated this aspect into the limitations section.
- Conclusions: What recommendations would the authors have for social media to help youth identify, address/cope with the emotional distress from becoming mental illness symptoms, since youth turn to social media so much. Does social media have a role in helping identifying youth in need of help, who are more likely to report negative emotional reactions?
Authors: we thank the reviewer for addressing the role of social media, which may be useful in identifying subjects at risk. Given the importance of understanding the direction of the association reported, we added the following sentence: Further studies are needed to understand the role of social media in identifying youth reporting negative emotional reactions, as they may benefit from supportive interventions.
Additionally, do the the authors feel that their findings indicate importance of helping youth make connection as a way out of anger , sadness ,etc. And, therefore, they would recommend that research on school-based preventative interventions should examine whether those intervention promote connection and connectedness, and what impact that has as protective factor against anger, sadness, boredom/emptiness, and anxiety as reported in this study. Should research and practice be recommended on gender specific programming (such as all girls or boys programming) as opposed to mixed-sex group programming given the findings of the study. Or would it be reasonable to recommend that, overall, social isolation is the major culprit here, and must be understood with new research and combatted with new programming?
Authors: We thank the reviewer for these additional comments. We highlighted in the text the protective factors found, as this might inform future post-pandemic preventive strategies and school-based interventions. Gender differences might be taken into account as well. We agree that overall, addressing social connections, especially among girls, might be considered a key aim of interventions in adolescents. Thus, we mentioned the importance of social connection within the conclusive section.